# Inorganic-organic competitive coating strategy derived uniform hollow gradient-structured ferroferric oxide-carbon nanospheres for ultra-fast and long-term lithium-ion battery

Yuan Xia[1], Tiancong Zhao[1], Xiaohang Zhu[1], Yujuan Zhao[1], Haili He[1], Chin-te Hung[1], Xingmiao Zhang[1], Yan Chen[1], Xinlei Tang[1], Jinxiu Wang[1], Wei Li [1✉] & Dongyuan Zhao [1✉]

The gradient-structure is ideal nanostructure for conversion-type anodes with drastic volume change. Here, we demonstrate an inorganic-organic competitive coating strategy for constructing gradient-structured ferroferric oxide-carbon nanospheres, in which the deposition of ferroferric oxide nanoparticles and polymerization of carbonaceous species are competitive and well controlled by the reaction thermodynamics. The synthesized gradient-structure with a uniform size of ~420 nm consists of the ferroferric oxide nanoparticles (4–8 nm) in carbon matrix, which are aggregated into the inner layer (~15 nm) with high-to-low component distribution from inside to out, and an amorphous carbon layer (~20 nm). As an anode material, the volume change of the gradient-structured ferroferric oxide-carbon nanospheres can be limited to ~22% with ~7% radial expansion, thus resulting in stable reversible specific capacities of ~750 mAh g$^{-1}$ after ultra-long cycling of 10,000 cycles under ultra-fast rate of 10 A g$^{-1}$. This unique inorganic-organic competitive coating strategy bring inspiration for nanostructure design of functional materials in energy storage.

[1] Department of Chemistry, Shanghai Key Lab of Molecular Catalysis and Innovative Materials, and Laboratory of Advanced Materials, Fudan University, Shanghai, P. R. China. ✉email: weilichem@fudan.edu.cn; dyzhao@fudan.edu.cn

Rechargeable lithium-ion batteries (LIBs) have been recognized as the most important power supply for portable electronics and electric vehicles[1–3]. Recently, there is an ever-growing demand to develop next-generation LIBs with high energy density, long cycling life, and low cost[4–6]. The key issue is creating high-capacity and ultra-stable electrode materials. In this regard, transition metal oxides and silicon have been regarded as the most promising candidates for the next-generation LIBs because of their high theoretical capacities[7–10]. Nevertheless, the increased specific capacity is generally accompanied by many challenges based on their conversion-type lithium-storage mechanism, such as large volume change[11–13], and low electronic/ionic conductivity[14–16]. Actually, the drastic volume changes are more lethal than other problems in the case of fast charging and discharging, which leads to unstable solid-electrolyte interphase (SEI) film, severe electrode pulverization, and loss of electrical contact, consequently, thus resulting in rapid capacity fading and even the safety problem of the battery[17,18].

Over the past decade, various nanostructure designs have been proposed to tackle the problems associated with the transition metal oxides and silicon anodes[19–25]. Reducing the active material size from the micrometer to the nanoscale regime is the most direct strategy to relieve the stress caused by volume change. In fact, a wide range of nanostructures have been developed and intensively investigated such as nanoparticles[26,27], 1D nanorods/nanowires[28,29], 2D nanosheets[30,31], porous structures[32–34], hollow[35,36] and hierarchical structures[37–39]. But the nanostructuring active materials inevitably increases the electrode–electrolyte contact area, thus, increases the dissolution and formation of SEI film, resulting in a low Coulombic efficiency and poor cycle life. Surface modification to form core-shell structures is a useful strategy for stabilizing SEI film by minimizing the electrode/electrolyte interfacial side reaction, increasing the electronic conductivity, and thus improving the lifetime of active materials[40–42]. Unfortunately, it is greatly challenging to construct a stable coating layer to retain the SEI film because the huge volume change of active cores is associated with Li$^+$-ion insertion/extraction process. The promising strategy to address this issue is to reserve space for volume change to form yolk-shell structure, in which the interior void space buffers the drastic volume change without cracking the overall electrode, and the outer shells can help to stabilize the SEI film. Nevertheless, in this kind of structures, the active cores are generally movable and vulnerable, making it difficult to achieve robust electronic connections and protective effect with the outer conductive shells, furthermore, the yolk void spaces could not be fully utilized during the large volume expansion and change, as a result, leading to unsatisfactory rate and cycling performances. Despite progresses have been achieved, it remains great challenge to develop a reasonable structure for substantially improving the battery performance to an ultrafast and long-life level. Gradient-structure, in which the active and modified materials combine in a sufficient contact way at the nanoscale, can drastically increase interaction areas, gradually release the stress caused by the volume change, therefore is an ideal structure for improving the stability of electrodes during ultrafast charging and discharging. However, constructing such gradient-structure has not been successful yet until now because it is difficult to control gradient component distribution by variable deposition reaction between different materials.

Herein, inspired by the structure of coconut, we design a hollow gradient-structured ferroferric oxide-carbon (HG-Fe$_3$O$_4$@C) nanospheres for ultrafast, long-term lithium-ion battery anodes. Using metallorganic compound ferrocene as the sole source for carbon and iron, we demonstrate a unique inorganic–organic competitive coating strategy for constructing a uniform gradient-structured Fe$_3$O$_4$@C shell on the surface of colloidal silica nanospheres, in which the deposition of Fe$_3$O$_4$ nanoparticles and polymeric carbonaceous species are competing gradually and well-controlled by the reaction thermodynamics. After removal of the silica spherical cores, the as-obtained nanospheres show a unique hollow gradient-structure with highly uniform particle size of ~420 nm. Where the Fe$_3$O$_4$ nanoparticles (4–8 nm) conformally coated by ultrathin conductive graphitic carbon are aggregated into the inner layer of carbonaceous matrix with high-to-low component distribution from inside to out (15 nm), which is encapsulated by an amorphous carbon layer (~20 nm in thickness). When being used as anodes for lithium-ion storage, the as-obtained uniform gradient-structured Fe$_3$O$_4$@C nanospheres deliver an impressive reversible capacity of ~750 mAh g$^{-1}$ with a Coulombic efficiency as high as ~99.0% even after 10,000 cycles at a high current density of 10 A g$^{-1}$, which is immensely higher than that of yolk-shell structured Fe$_3$O$_4$@C (350 mAh g$^{-1}$) and hollow hybrid structured Fe$_3$O$_4$@C (330 mAh g$^{-1}$) nanospheres. Even at an ultrafast rate of 20 A g$^{-1}$, significantly high capacities of ~500 mAh g$^{-1}$ can be retained after 10,000 cycles. We believe that the unique inorganic–organic competitive strategy is easy, reproducible, general, and can bring inspiration for nanostructuring and composites design of next-generation high-performance anodes.

## Results

**Morphological and structural characterization**. The synthesis procedure of the hollow gradient-structured Fe$_3$O$_4$@C nanospheres is illustrated through the inorganic–organic competitive coating strategy (Fig. 1). In this synthesis, ferrocene, a metallorganic compound, is used as the sole source both for carbon and iron to directly coat Fe$_3$O$_4$@C shells on the surface of colloidal silica nanospheres. Then, a high-to-low gradient Fe$_3$O$_4$@C component distribution can be obtained via a simple solvothermal process. Field-emission scanning electron microscopy (FESEM) images show that the as-made core-shell nanospheres are highly uniform with a diameter of ~420 nm (Supplementary Fig. 1). Compared with that for the initial colloidal silica cores (~350 nm), the diameter of the gradient-structured nanospheres increases, clearly indicating that the coating thickness is about 70 nm. After the calcination and selective removal of the silica cores, the obtained hollow gradient Fe$_3$O$_4$@C nanospheres with an inner void of ~350 nm show monodispersed size distribution of ~420 nm with inconspicuous shrinking (Fig. 2A–E). The magnified TEM images of a single Fe$_3$O$_4$@C nanosphere disclose the unique gradient-structure (Fig. 2D, E). Where the Fe$_3$O$_4$ nanoparticles (4–8 nm in size) in carbonaceous matrix are aggregated into the inner layer with high-to-low component distribution from inside to out, which is encapsulated by an outer amorphous carbon layer (~20 nm) without Fe$_3$O$_4$. The hollow structure can also be confirmed by the scanning electron microscope (SEM) images of a broken nanosphere (Fig. 2C). High-resolution TEM (HRTEM) images demonstrate that the Fe$_3$O$_4$ nanoparticles in the inner mesoporous layer are highly crystallized with a relatively uniform particle size of 4–8 nm and conformally wrapped by few graphitic carbon layers (Fig. 2F and Supplementary Fig. 2). Moreover, bright voids can be clearly observed between the Fe$_3$O$_4$ nanoparticles, which are the characteristic of mesoporous structures. Selective etching method is used to further confirm the gradient-structure of the obtained nanospheres. After etching by HF solution to remove the Fe$_3$O$_4$ nanoparticles from the carbonaceous matrix, the inner surface of the obtained hollow carbon nanospheres is pitted with craters (Supplementary Fig. 3), clearly indicating that the Fe$_3$O$_4$ nanoparticles are gradient-embedded in the inner wall of amorphous carbon layers. In contrast, when

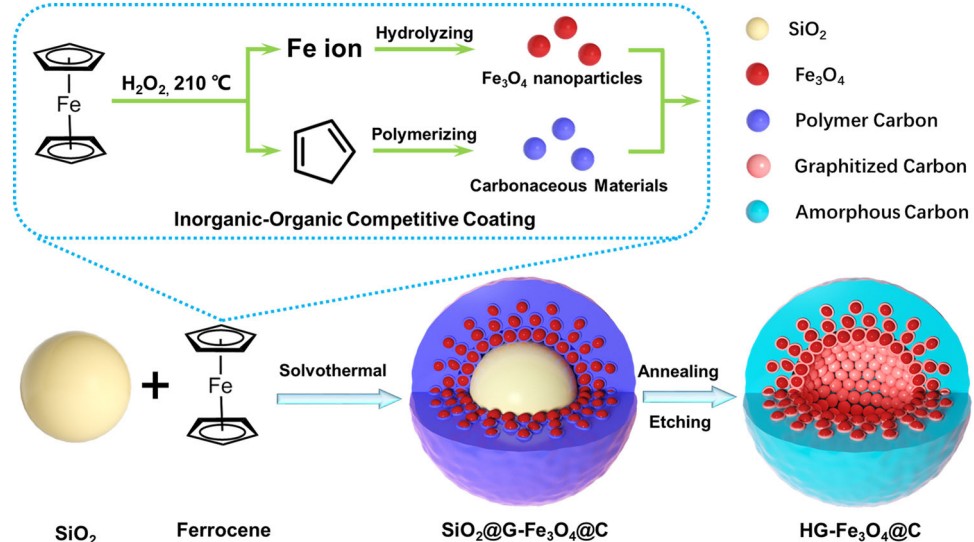

**Fig. 1 Schematic diagram for the synthesis of the hollow gradient-structured Fe$_3$O$_4$@C nanospheres via an inorganic-organic competitive coating strategy.** The colloidal SiO$_2$ nanospheres can be used as a sacrificial template for the hollow structure. Metallorganic compound, Ferrocene is selected as a sole source for both iron and carbon.

etching the as-made SiO$_2$@Fe$_3$O$_4$@C nanospheres with 1.0 M NaOH to remove the polymeric carbonaceous layers or matrixes, the hollow hybrid Fe$_3$O$_4$/C nanospheres without thick carbon outer layer can be obtained, which further proves the gradient distribution of carbon and Fe$_3$O$_4$ nanoparticles in the nanospheres (Supplementary Fig. 4). The selected-area electron diffraction (SAED) characterization reveals the high crystallization of Fe$_3$O$_4$ nanoparticles in the hollow gradient-structure (Fig. 2G). The scanning TEM (STEM) and corresponding EDS mapping images show that the hollow profiles of Fe element are well coincided with O but smaller than that of C, indicating that the gradient distribution Fe$_3$O$_4$ nanoparticles is encapsulated by an outer amorphous carbon layer (Fig. 2H–K). From the in situ TEM characterization based on the electron tomography and spinning projection technology (Video 1 and 2), it is further demonstrated that the distinct gradient embedment of Fe$_3$O$_4$ nanoparticles within the carbon shells, and the unique "occlusion"-like schemes between the two materials.

X-ray diffraction (XRD) patterns (Supplementary Fig. 5A) of the gradient-structured Fe$_3$O$_4$@C nanospheres show that all diffraction peaks can be well assigned to pure magnetite Fe$_3$O$_4$ (JCPDS card no.11-0614). The average crystallite sizes of the Fe$_3$O$_4$ can be calculated to ~5.3 nm by Debye–Scherrer formula. This value is well-matched with the TEM data of (4–8 nm). The G band of the carbon in the Raman spectra (Supplementary Fig. 5B) shift to a higher wave number of 1613 cm$^{-1}$ compared with that of the graphite single crystal (1575 cm$^{-1}$), indicating that a large amount of disordered graphite-like carbon in the gradient-structured Fe$_3$O$_4$@C nanospheres. The carbon content in the gradient-structured Fe$_3$O$_4$@C nanospheres can be estimated to be about 23.8% based on the TGA data (Supplementary Figs. 5C, 6, 7). The N$_2$ adsorption/desorption isotherms of the gradient-structured Fe$_3$O$_4$@C nanospheres show a typical type III feature (Supplementary Fig. 5D), indicating the microporosity of this mesoporous structure aggregated by nanoparticles is non-significant. The micropore volumes determined from N$_2$ adsorption and CO$_2$ adsorption (Supplementary Fig. 5E) are 0.005 and 0.007 cm$^3$ g$^{-1}$, respectively, being negligible compared with the total pore volume (0.32 cm$^3$ g$^{-1}$). The BET surface area of the gradient-structured Fe$_3$O$_4$@C nanospheres is about 141 m$^2$ g$^{-1}$. Pore size distribution obtained from Barrett–Joyner–Halenda

model indicates the presence of mesopore with pore size at around 4 nm (Supplementary Fig. 5F).

**The formation process of gradient-structures and its versatility.** The temperature- and time-dependent experiments were carried out to examine the deposition of Fe$_3$O$_4$ and amorphous carbonaceous (Supplementary Figs. 8, 9). When the reaction temperature is set as 180 °C, a pure carbon layer without visible Fe$_3$O$_4$ nanoparticles is coated on the surface of the colloid SiO$_2$ nanospheres (Supplementary Fig. 8A, E, I). Corresponding XRD pattern further confirms that the Fe$_3$O$_4$ nanoparticles cannot be formed at such low temperature (Supplementary Fig. 8M). Increasing the reaction temperature to 190 °C, Fe$_3$O$_4$ nanoparticles can be observed in a form of aggregations, which adhere to the surface of a pure carbon layer (Supplementary Fig. 8B, F, J). Interestingly, when the temperature increases to 200 °C, the Fe$_3$O$_4$ nanoparticles are embedded in carbonaceous matrix at the inner of the coating-layer to form a gradient high-to-low component distribution from inside to out (Supplementary Fig. 8C, G, K). Further increasing the reaction temperature, Fe$_3$O$_4$ nanoparticles are locally concentrated at the inner walls of the coating-layer, and no Fe$_3$O$_4$ particles can be observed within the outer carbon walls (Supplementary Fig. 8D, H, L). In this case, it can clearly be seen that at the beginning of deposition process, the Fe$_3$O$_4$ nanoparticles covered by ultrathin carbon layers are rapidly deposited on the colloid SiO$_2$ cores (Supplementary Fig. 9A, B). With the reaction, the thickness of the Fe$_3$O$_4$-rich layers is increased, which is then encapsulated by an out carbon wall after 24 h (Supplementary Fig. 9C, D). XRD patterns show that the diffraction intensities of the magnetite nanoparticles increase with the reaction temperature, suggesting that the deposition of Fe$_3$O$_4$ nanoparticles is determined on thermodynamics (Supplementary Fig. 8M). From another point of view, the carbon layers with a similar thickness can be formed from 180 to 210 °C, indicating that the coating rate slightly increases at the controlled temperatures (Supplementary Fig. 10).

It is worth noting that our inorganic–organic competitive coating strategy is simple and versatile to synthesize uniform gradient-structured Fe$_3$O$_4$@C nanospheres with controllable particle size from 150 to 500 nm and shell thicknesses ranging

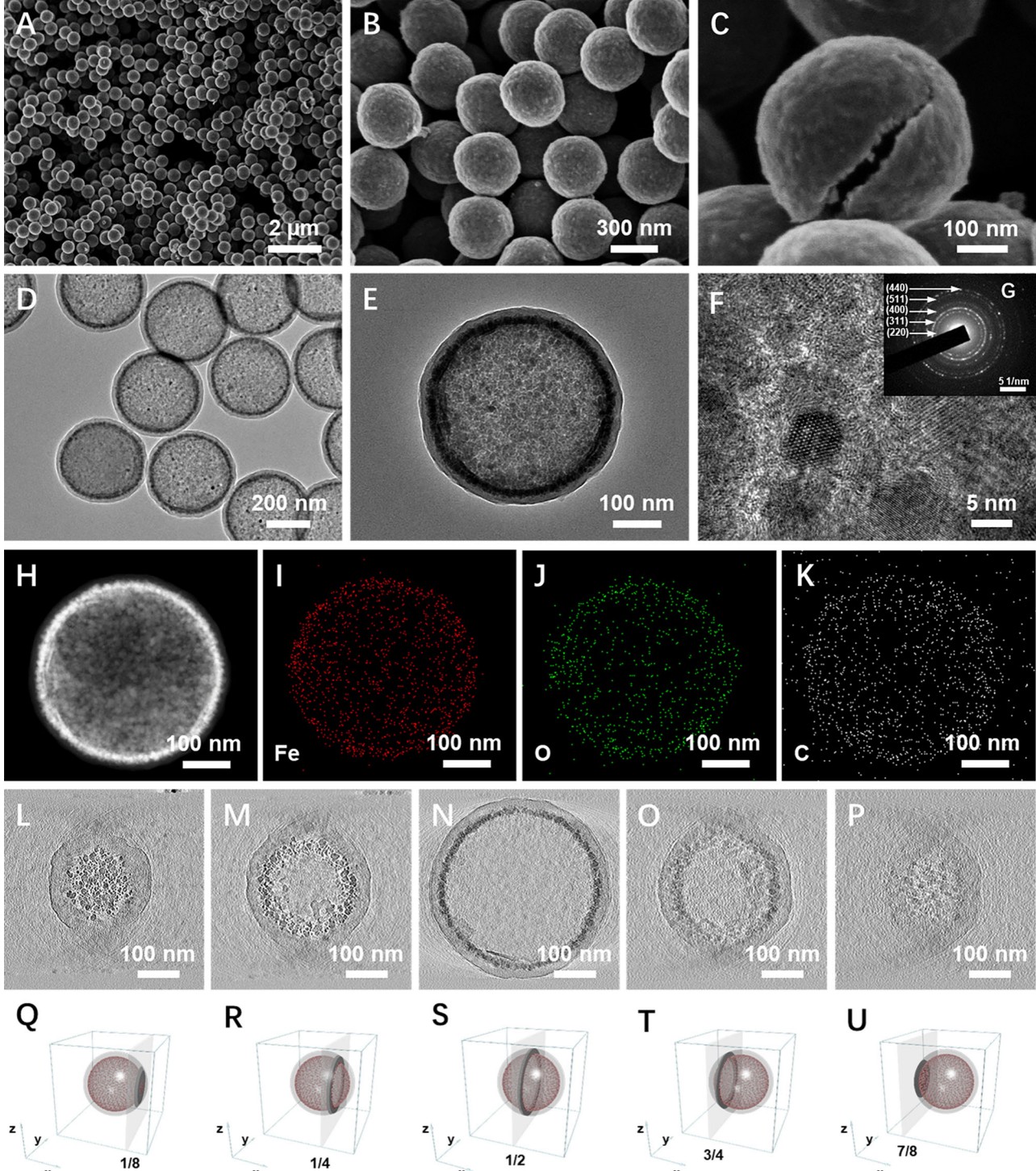

**Fig. 2 Morphology characterization of the hollow gradient-structured Fe₃O₄@C nanospheres. A**, **B**, **C** SEM and **D**, **E** TEM images of the gradient-structured Fe₃O₄@C nanospheres; **F** HRTEM image of the gradient-structured Fe₃O₄@C nanospheres; The inset in (**F**) is the selective area electronic diffraction (SAED) pattern (**G**) of the sample gradient-structured Fe₃O₄@C nanospheres; **H** STEM, and **I**, **J**, **K** corresponding EDS mapping of Fe, O, and C elements distribution in the typical nanosphere; **L–P** TEM images of different slice depths of the sample gradient-structured Fe₃O₄@C nanosphere and **Q–U** corresponding section schematic.

from 20 to 80 nm (Supplementary Figs. 11, 12). More interestingly, when the concentration of colloidal silica nanospheres increases four times, the nucleated Fe₃O₄ nanocrystals are not enough to cover the whole surface of the templates. Thus, a strawberry-like growth of gradient-structured Fe₃O₄@C shells can be obtained (Supplementary Figs. 13, 14), further indicating that

the inorganic–organic competitive and interface-induced deposition process. Moreover, this competitive coating strategy is versatile, various functional cores, such as core-shell Fe₃O₄@SiO₂ nanospheres and core-shell SiO₂@G-Fe₃O₄@C (SiO₂ coated by gradient-structured Fe₃O₄@C layer) nanospheres, can be obtained with hollow structures, leading to the formation of

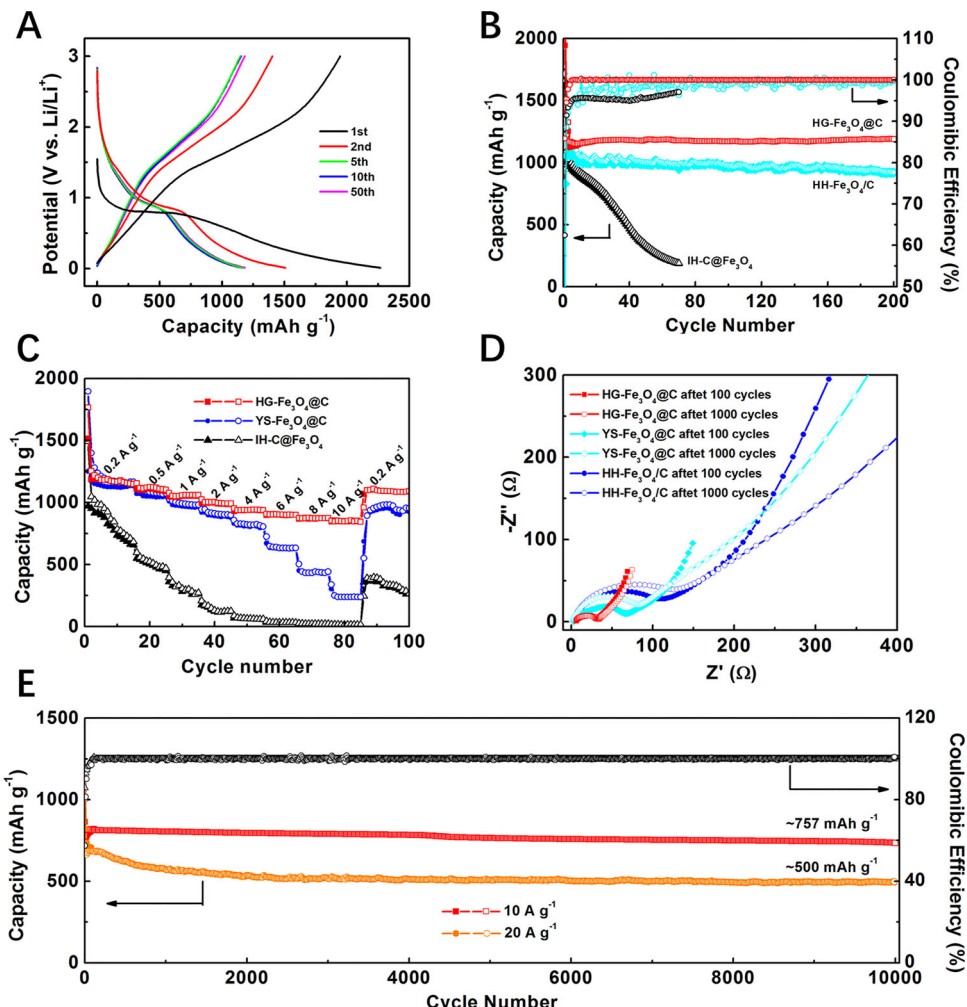

**Fig. 3 Electrochemical characterizations of the hollow gradient-structured Fe₃O₄@C nanospheres (HG-Fe₃O₄@C) and the control samples application in lithium ion batteries. A** Charge-discharge curves in the voltage range from 0.05 to 3.00 V; **B** Cycling performances at a current density of 0.2 A g⁻¹; **C** Rate performances at various current rates from 0.2 to 0.5, 1, 2, 4, 6, 8, 10 A g⁻¹; **D** Nyquist plots between 100 to 0.05 kHz; **E** Cycling performances under a high current density of 10 and 20 A g⁻¹. Hollow hybrid Fe₃O₄@C (HH-Fe₃O₄/C), yolk-shell Fe₃O₄@C (YS-Fe₃O₄@C), and island-type C@Fe₃O₄ (HI-C@Fe₃O₄) nanospheres were prepared for comparison.

yolk-shell structures with hybrid gradient-structured shells and well-defined interior void spaces (Supplementary Fig. 15). We also show that the inorganic–organic competitive coating strategy can further be extended to prepare uniform $SiO_2@TiO_2@C$ and $SiO_2@NiO@C$ core-shell structures by using titanocene and nickelocene as the reactants, respectively (Supplementary Figs. 16, 17). Therefore, our coating method is versatile and robust for synthesizing core-shell and hollow structures with multiple functions and gradient-structured interface.

**Electrochemical performance.** The lithium-storage performances of the gradient-structured $Fe_3O_4@C$ nanospheres as an anode material were tested in the typical half-cell configuration. All capacity values reported here are based on the total mass of $Fe_3O_4$ and C in the nanospheres. The first three CV curves of the hollow gradient-structured $Fe_3O_4@C$ electrode at a scan rate of 0.2 mV s⁻¹ from 0.0 and 3.0 V (vs Li/Li⁺) show a characteristic of typical metal oxide anodes (Supplementary Fig. 18). In the first cathodic scan, the broad reduction peak at 0.6 V can be assigned to the SEI formation and irreversible reactions, which disappear in the subsequent cycles. A connected wide peak in the anodic scan can be attributed to the oxidation of $Fe^0$ to $Fe^{3+}$. From the second cycle, two coupled redox

peaks appeared in the CV curves, indicating the reversible electrochemical behavior of the gradient-structured $Fe_3O_4$ nanospheres. The charging and discharging curves of the gradient-structured $Fe_3O_4@C$ electrode (Fig. 3A) also exhibit typical electrochemical features of $Fe_3O_4$ anodes with a negligible change between the fifth and 50th cycles. The lithium-storage characteristic of gradient-structured $Fe_3O_4@C$ nanospheres were also tested at low current density of 0.1 C (1 C = 926 mA g⁻¹, Supplementary Fig. 19). One clear declining plateau is observed in all discharging and charging curves, respectively, implying a wide potential window of the gradient-structured $Fe_3O_4@C$ nanospheres as a lithium ion battery anode.

To evaluate the structure-performance relationship, three commonly used anode without gradient-structures, including hollow hybrid $Fe_3O_4@C$ (HH-$Fe_3O_4$/C), yolk-shell $Fe_3O_4@C$ (YS-$Fe_3O_4@C$), and hollow island-type $C@Fe_3O_4$ (HI-$C@Fe_3O_4$) nanospheres, were prepared for comparison (Supplementary Figs. 20, 21). When charging and discharging in a low current density of 0.2 A g⁻¹ (Fig. 3B), the gradient-structured $Fe_3O_4@C$ electrode demonstrates an excellent specific capacity of 1200 mAh g⁻¹ with a Coulombic efficiency of ~99.8%, which is immensely higher than that of hollow island-type $C@Fe_3O_4$ (190 mAh g⁻¹) and hollow hybrid $Fe_3O_4@C$ (990 mAh g⁻¹, 98.8%).

Ex-situ TEM images (Supplementary Fig. 22) demonstrates the morphology of the typical hollow gradient-structured $Fe_3O_4$@C nanosphere at different charging and discharging states during the fifth cycle. The diameter of the hollow nanosphere is about 415 nm at the beginning, and increased to 426 and 449 nm at the half and full discharging, respectively. In contrast, the corresponding void size decreases from 354 to 348 and 345 nm gradually for the buffering volumetric expansion. About 7% radial expansion of the gradient-structured $Fe_3O_4$@C nanospheres can be evaluated from the full charging to full discharging state, which is very close to that of commercial graphite anode (~6% expansion in the direction perpendicular to the (002) crystal surface), and considerably lower than the theoretical value of $Fe_3O_4$ particles (80% volume expansion, ~22% radial expansion). When being charged, the structure dimension can be restored to its original state. Notably, the well-designed hollow gradient-structure can be perfectly retained at different states of charging and discharging, even after 100 or 10000 cycles at the different current densities of 0.2 or 10 A g$^{-1}$ (Supplementary Figs. 23, 24).

As the current density increases from 0.2 to 0.5, 1, 2, 4, 6, 8, and 10 A g$^{-1}$ for fast charging and discharging (Fig. 3C), the reversible capacities of the gradient-structured $Fe_3O_4$@C electrode is decreased from 1200 to 1120, 1060, 990, 940, 900, 870, and 850 mAh g$^{-1}$, respectively. When the rate returns to 0.2 A g$^{-1}$, the capacities of the gradient-structured $Fe_3O_4$@C electrode comes back to ~1180 mAh g$^{-1}$, indicating an excellent rate reversibility based on the unique gradient-structure. As a comparison, the yolk-shell $Fe_3O_4$@C electrode with a similar core-shell structure but without gradient $Fe_3O_4$ distribution can deliver comparable capacities with gradient-structured $Fe_3O_4$@C at low current densities. However, when the current densities exceed 1 A g$^{-1}$, it shows capacities of 990, 910, 820, 630, 430, and 240 mAh g$^{-1}$ at 1, 2, 4, 6, 8, and 10 A g$^{-1}$, respectively. In fact, the cycle life and capacities of the gradient-structured $Fe_3O_4$@C nanospheres are much better than that of the previous works under ultrafast charge and discharge conditions (Supplementary Table 1). Importantly, the gradient-structured $Fe_3O_4$@C electrode can stand stable for a long term under fast charging and discharging. After 10,000 cycles, it can deliver capacity as high as ~750 mAh g$^{-1}$ with a Coulombic efficiency closed to 99.0% at 10 A g$^{-1}$ (Fig. 3E). This value is more than two times higher than that of the yolk-shell $Fe_3O_4$@C (350 mAh g$^{-1}$, Supplementary Fig. 25) and hollow hybrid $Fe_3O_4$@C (330 mAh g$^{-1}$, Supplementary Fig. 26) electrodes. Even at 20 A g$^{-1}$, the capacities of the gradient-structured $Fe_3O_4$@C electrode are still as high as 500 mAh g$^{-1}$ after 10,000 cycles with a capacity fading <0.003% per cycle, implying that our electrode can afford ultrafast charging and discharging for a long time. The ex-situ SEM images (Supplementary Fig. 27) show that thin, stable, and spatially confined SEI films are formed in the gradient-structured $Fe_3O_4$@C electrode. All of the gradient-structured $Fe_3O_4$@C nanospheres were retained intact very well on the surface of the electrode, indicating the excellent structural and thermal stability of gradient-structured $Fe_3O_4$@C nanospheres. As a contrast, due to the constraints of the mass/charge transfer, a thicker SEI film is coagulated on the yolk-shell $Fe_3O_4$@C electrode (Supplementary Fig. 28), which not only increases the interface resistance of the electrode but also leads to the fluctuating Coulombic efficiency (between 96.0 and 100.5%) and inferior capacities. The morphology of the hollow hybrid $Fe_3O_4$/C (HH-$Fe_3O_4$/C) electrode is almost entirely damaged after 1000 cycles due to the spallation of SEI film caused by the large volume change of metal oxide $Fe_3O_4$ (Supplementary Fig. 29). Thus, the capacities of the hollow hybrid $Fe_3O_4$@C-$Fe_3O_4$/C electrode gradually fade from 650 to 330 mAh g$^{-1}$ with low Coulombic efficiencies during 10,000 cycles. Those morphological changes can be further confirmed by the EIS results (Fig. 3D). The

gradient-structured $Fe_3O_4$@C electrode shows a constant interface resistance of ~35 Ω after 100 or 1000 cycles, which is much lower than that of the yolk-shell structured $Fe_3O_4$@C (62 Ω after 100 cycles and 83 Ω after 1000 cycles) and the hollow hybrid $Fe_3O_4$/C (110 Ω after 100 cycles, and 135 Ω after 1000 cycles). It further suggests that the gradient-structure is able to gradually release the stress caused by drastic volume change during fast charging and discharging.

Considering the fact that asymmetric charging and discharging are commonly applied in practical applications, it is important to evaluate the battery performances of the gradient-structured $Fe_3O_4$@C electrodes at different charge and discharge current densities. When charging at 0.2 A g$^{-1}$ and discharging at 10 A g$^{-1}$, a high capacity of ~900 mAh g$^{-1}$ and Coulombic efficiency of ~97% can be obtained with negligible fading after 200 cycles. The result suggests that Li$^+$ diffusion thermodynamics and dynamics performances of at a high current are comparable with those at a low current in the solid phase (Supplementary Fig. 30). On the other hand, the cyclic stabilities of the gradient-structured $Fe_3O_4$@C can be maintained well with high loading density at 10 A g$^{-1}$. With the increasing, the mass loading to 5, 10, and 20 mg/cm$^2$, the capacities of gradient-structured $Fe_3O_4$@C nanospheres are decreasing to 727, 635, and 590 mAh g$^{-1}$, respectively, based on the thickness increasing of the electrode (Supplementary Fig. 31).

## Discussion

**Inorganic-organic competitive coating strategy**. We propose an inorganic–organic competitive coating and deposition process for the formation of the gradient-structured $Fe_3O_4$@C shells (Fig. 4). At the beginning of the solvothermal reaction, the metallorganic ferrocene can be gradually hydrolyzed into iron ions and carbonaceous species by $H_2O_2$ (Fig. 4A). Then, the iron ions would be further hydrolyzed into Fe oxides and carbonaceous species could be polymerized into an amorphous carbon layer under the solvothermal condition (Fig. 4B). Thus, two competitive deposition reactions are occurred and much dependent on the thermodynamics for the controllable growth of unique gradient-structured $Fe_3O_4$@C shells (Fig. 4C). At a relatively low temperature (180 °C), iron ions cannot fully be dissociated from coordination of both cyclopentadienes and hydrolyzed, thus iron oxide species cannot form under this condition. In contrast, some cyclopentadiene species partially decomposed from ferrocene can be polymerized into amorphous carbonaceous species, which can be coated on the surface of the colloidal $SiO_2$ cores driven by the affinity interaction between the abundant Si-OH groups and hydroxy/carboxyl groups. Thus, uniform $SiO_2$@C core-shell structures are obtained. Increasing the solvothermal temperature (190 °C) can slightly accelerate the polymerization reaction of carbonaceous species into a thicker layer. Moreover, the nucleation and growth of $Fe_3O_4$ nanoparticles can be triggered by the cross-linking and hydrolysis of iron ion species decomposed from ferrocene. Simultaneously, an ultrathin carbon layer is formed on the surface of $SiO_2$@C nanospheres, forming an island-like morphology. When applying a high solvothermal temperature (200 °C), the thermodynamics of the cross-linking, nucleation and growth of $Fe_3O_4$ nanocrystals is boosted, which is faster than that of the carbonaceous polymerization process. Therefore, $Fe_3O_4$ nanocrystals are first formed, then the amorphous carbonaceous species are coated on their surface due to the high surface energy of small nanocrystals, leading to the fast formation of $Fe_3O_4$@C nanocrystals. Subsequently, the nanocrystals are self-assembled on the surface of colloidal $SiO_2$ cores into the "reinforced concrete" inner layers to lower the total system energy. Furthermore, the amorphous carbonaceous layers are deposited outside with the exhaustion of iron species with

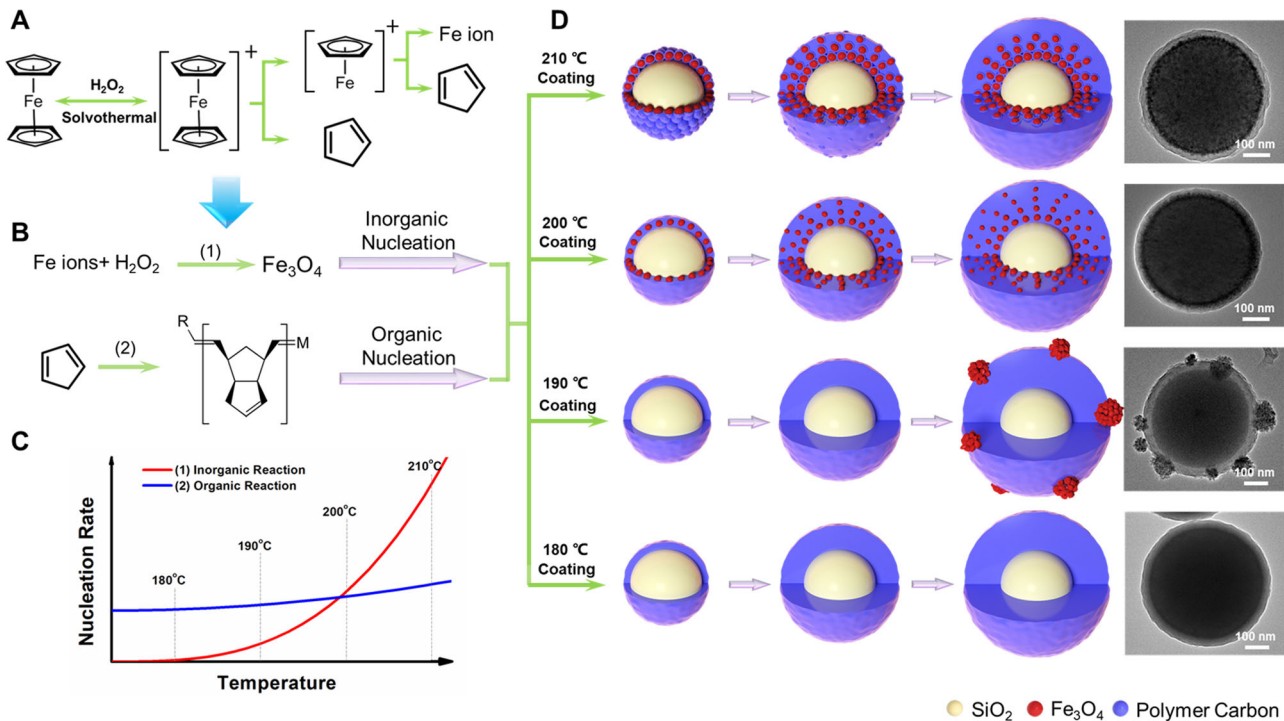

**Inorganic-Organic Competitive Coating Strategy**

**Fig. 4 Schematic representation of the inorganic–organic competitive coating strategy. A** The metallorganic ferrocene was gradually hydrolyzed into iron ions and cyclopentadienes; **B** Iron ions were further hydrolyzed into hydrated iron oxides (inorganic nucleation), and cyclopentadienes were oxidized and polymerized into amorphous carbonaceous species (organic nucleation); **C** Schematic diagram for nucleation rate between iron oxides and amorphous carbonaceous species; **D** Model diagram of competitive coating process by solvothermal reaction temperature of 210, 200, 190, and 180 °C.

pallid domain of $Fe_3O_4$@C nanocrystals, forming a gradient distribution coating-layer. Further increasing the temperature leads to the rapid nucleation and growth of $Fe_3O_4$ nanocrystals (210 °C). As a result, the most $Fe_3O_4$ nanocrystals are locally concentrated at the inner walls of the coating-layer. With the subsequent overgrowth of amorphous carbonaceous layers, the coconut-like gradient-structured $Fe_3O_4$@C shells are eventually formed. Most of $Fe_3O_4$ nanoparticles with a size of 4–8 nm are aggregated into the inner layer (~10 nm) and form a gradient high-to-low component distribution from inside to out (15 nm). Interestingly, after the carbonization at 600 °C, the carbon layer around the $Fe_3O_4$ nanoparticles can be easily graphitized due to the catalytic effect of iron ions, forming a graphitized carbon coating with few layers. In contrast, the thick carbon layer is still amorphous structures. Therefore, such inorganic–organic competitive coating strategy can be effectively applied to construct unique gradient-structure shells through well regulating the competitive reaction thermodynamics (Fig. 4D).

**Structure activity relationship**. One of the advantages of our gradient-structure is ensuring a sufficient "occlusion"-like contact between the active material and conductive shell, which can gradually relieve the stress concentration caused by drastic volume change, thus retaining the completeness and stability of the whole structure during ultrafast charging and discharging (Fig. 5). At the half discharge/charge state, the diameter of the gradient-structured $Fe_3O_4$@C nanospheres increases by only about 2%, indicating that the inner mesopores of the gradient-structure present a sustained-buffering effect on the drastic volume change of the active $Fe_3O_4$. Moreover, the gradient-structure gradually relies on the void space to alleviate the volume

change in the following discharging process. About 7% of the diameter increase for the gradient-structured $Fe_3O_4$@C nano-spheres can be confirmed in comparison to the full charge and full discharge state, which is similar to the expansion coefficient of graphite anodes (6%), and doesn't cause the burst of SEI film on the surface. Although a certain amount of volumetric energy density is sacrificed, the mass energy density is improved qualitatively. Thus, the morphology of SEI layer and the whole electrode can be retained after hundreds of cycles, suggesting an ultra-long life of the gradient-structured $Fe_3O_4$@C eletrodes with the gradient-structure. In addition, the gradient-structure design affords other advantages for ultrafast and long-life performance (Fig. 5). In fact, each nanocrystallized $Fe_3O_4$ particle is con-formally encapsulated by ultrathin graphitic carbon layers and then embedded in the inner wall of hollow carbon nanospheres, thus, resulting a very good ohm contact between active substance and modified conductive material. Furthermore, the ultrathin graphitic carbon layer ensures active particles independent of each other, thus prevent agglomeration of secondary particles during the charge and discharge cycling. As shown, even in full discharge state or after multiple cycles, $Fe_3O_4$ particles retain the initial encapsulated structure. Moreover, the inner active layer in the gradient-structure is encapsulated by an out amorphous carbon inactive layer, which enables SEI deposited on the outer surface of whole structures, thus spatially confined SEI formation (Supplementary Fig. 26).

In summary, we report an inorganic–organic competitive coating strategy for the synthesis of unique hollow gradient-structure of $Fe_3O_4$@C nanospheres by using metalorganic compound ferrocene as a sole precursor. Using colloidal silica as a template, the competitive coating strategy can be achieved by one-step solvothermal reaction. The results show that the two

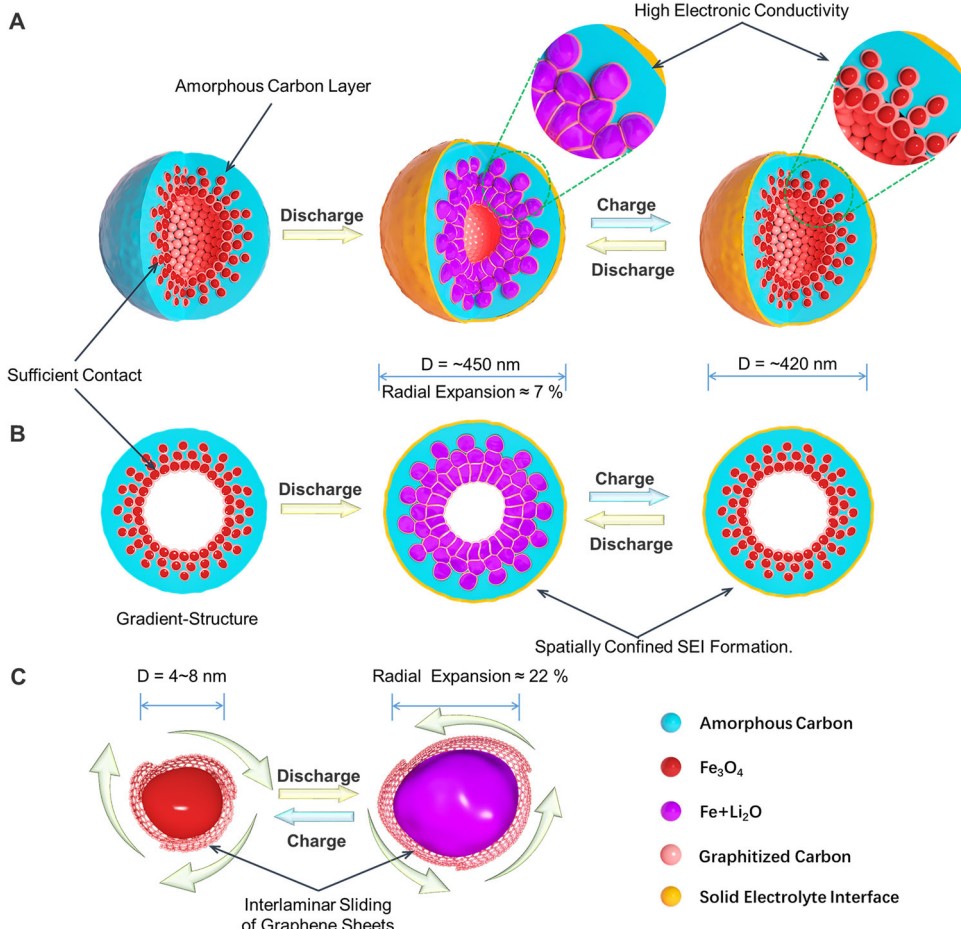

**Fig. 5 Schematic illustrations showing the structural change of the hollow gradient-structured Fe₃O₄@C nanospheres during fast charging and discharging. A** Three-dimensional (3D) view for the structural change of the hollow gradient-structured $Fe_3O_4$@C nanospheres; **B** simplified 2D cross-section view for the structural change of the hollow gradient-structured $Fe_3O_4$@C nanospheres; and **C** volume change of single $Fe_3O_4$ nanoparticle. ~7% diameter expansion of the gradient-structured nanospheres can be evaluated from the full charging to full discharging state, which is considerably lower than the theoretical value of $Fe_3O_4$ particles (~22%). proving that the gradient-structure can gradually release the stress caused by the volume change.

competitive cross-linking and polymerization reactions between organic species and inorganic iron precursors from ferrocene can be effectively controlled by the thermodynamics for the formation of unique gradient-structured $Fe_3O_4$@C shells. In fact, the metallorganic ferrocene can be gradually hydrolyzed into iron ions and cyclopentadienes by $H_2O_2$, then, iron ions could be further hydrolyzed and cross-linked into hydrated iron oxide species, and cyclopentadienes could be oxidized and polymerized into carbonaceous layer. As a typical hollow gradient-structured nanosphere, $Fe_3O_4$ nanoparticles (4–8 nm) conformably coated by ultrathin conductive graphitic carbon are aggregated into the inner layer with high-to-low component distribution from inside to out (15 nm), which is encapsulated by an amorphous carbon layer (~20 nm). In fact, the particle size (from 150 to 500 nm) and shell thicknesses (from 20 to 80 nm) of the hollow gradient-structured nanospheres can be easily adjusted. We also show that this synthesis approach is versatile and can be easily extended to prepare other core-shell and hollow gradient structures, such as $SiO_2$@void@G-$Fe_3O_4$@C, $Fe_3O_4$@void@G-$Fe_3O_4$@C, $SiO_2$@G-NiO@C, and $SiO_2$@G-$TiO_2$@C nanospheres. When being used as an electrode, such unique gradient-structure can effectively relieve the stress concentration caused by drastic volume change, and demonstrate excellent stability during fast charging and

discharging. As a result, the hollow gradient-structured $Fe_3O_4$@C nanospheres exhibit a highly reversible capacity of ~750 mAh g⁻¹ after 10,000 cycles at a high current density of 10 A g⁻¹, which is more than twice as high as that of the well-known yolk-shell (350 mAh g⁻¹) and hollow hybrid structure (330 mAh g⁻¹) anodes. Even at the current density as high as 20 A g⁻¹, the capacity of the gradient-structured $Fe_3O_4$@C electrode is still as high as 500 mAh g⁻¹ during 10,000 cycles with a capacity fading of <0.003% per cycle. More importantly, the gradient-structured $Fe_3O_4$@C electrode retains stable capacities of about ~900 mAh g⁻¹ even when asymmetric charge and discharge (discharge at 10 A g⁻¹, charge at 0.2 A g⁻¹). We believe that the gradient-structure will bring a revolution in the field of ultrafast charge and discharge for LIBs. The inorganic–organic competitive coating strategy paves a way for the design and synthesis of core-shell and hollow structured materials for energy storage.

## Methods

**Synthesis of materials**. The hollow gradient-structured $Fe_3O_4$@C (HG-$Fe_3O_4$@C) nanospheres were prepared by the inorganic-organic competitive coating strategy. Uniform colloidal $SiO_2$ nanospheres with a particle size of ~350 nm were synthesized and selected as a template core (Supplementary Fig. 1a, b). In a typical process, 0.20 g of the colloidal $SiO_2$ nanospheres and 0.40 g of metallorgaonic compound ferrocene were added into 25 mL of acetone with stirring for 10 min.

Next, 2.0 mL of hydrogen peroxide ($H_2O_2$) solution (25%) was added into the mixture. After continually stirring for 30 min, the mixture was transferred into an autoclave and heated at 210 °C for 24 h. The products were collected by centrifugation and washed with ethanol for three times, then and dried at room temperature. The as-made samples were calcined at 600 °C in $N_2$ atmosphere for 1 h. Finally, the gradient-structured $Fe_3O_4$@C nanospheres were obtained by removing the colloidal $SiO_2$ cores with 1.0 M NaOH aqueous solution.

The inorganic-organic competitive coating strategy could be extended to prepare uniform gradient-structured $TiO_2$@C and NiO@C layers. In a typical process for the gradient-structured $TiO_2$@C layer, 0.20 g of the colloidal $SiO_2$ nanospheres and 0.25 g of titanocene were added into 25 mL of acetone with stirring for 10 min. Next, 0.20 mL of hydrogen peroxide solution (25%) was added into the mixture. After continually stirring for 30 min, the mixture was transferred into an autoclave and heated at 150 °C for 24 h. The products were collected by centrifugation and washed with ethanol for three times, then and dried at room temperature. In a typical process for the gradient-structured NiO@C layer, 0.20 g of the colloidal $SiO_2$ and 0.15 g of nickelocene were added into 25 mL of acetone with stirring for 10 min. Next, 0.10 mL of hydrogen peroxide solution (25%) was added into the mixture. After continually stirring for 30 min, the mixture was transferred into an autoclave and heated at 120 °C for 24 h. The products were collected by centrifugation and washed with ethanol for three times, then and dried at room temperature.

**Characterizations**. The morphologies of the samples were investigated by using FESEM (Zeiss Supra 4VP) and transmission electron microscopy (TEM, JEOL JEM-ARM200F, with acceleration voltage of 200 keV). The samples for TEM measurements were suspended in ethanol and supported onto a holey carbon film on a Cu gird. FESEM images were taken on a Hitachi S-4800 microscope. The crystalline microstructures of samples were characterized by using XRD diffractometer (Bruker D8 diffractometer) with a Cu Kα radiation source (λ = 0.15406 nm). $N_2$ adsorption-desorption isotherms were recorded on a Micromeritics 3Flex analyzer at the temperature of 77 K. The samples were degassed in a vacuum at 150 °C for 6 h and the Brunauer–Emmett–Teller (BET) method was utilized to calculate the specific surface areas and pore size. The micropore volumes were determined from Dubinin–Radushkevich (DR) equation with both $N_2$ adsorption at 77 K and $CO_2$ adsorption at 273 K. The total pore volume was determined from the $N_2$ adsorbed amount at $P/P_0 = 0.99$ according to the Gurvitch rule. Thermogravimetric analysis (TGA) curves were acquired on a STA 449 C thermobalance with a temperature ramp of 10 °C min$^{-1}$.

**Cell assembly and electrochemical measurements**. Electrochemical measurements were carried out in 2016 type button cells with pure lithium foil as both counter and reference electrode. The active materials (for example, the hollow gradient-structured $Fe_3O_4$@C nanospheres) were mixed with the binder (poly (vinylidene fluoride)) and conductive agent (acetylene black) at a weight ratio of 8:1:1 to form a slurry. The mixing slurry was homogeneously pasted on nickel foam and dried in a vacuum oven at 80 °C for 12 h. After a hot- (90 °C, 30 Mpa) and cold-pressing (30 °C, 50 Mpa), the as-prepared electrodes were used to assemble the battery. The different areal mass loading of 1, 5, 10, and 20 mg/cm$^2$ was fabricated to investigate the lithium storage performance of gradient-structured $Fe_3O_4$@C ($HG$-$Fe_3O_4$@C) nanospheres. The electrolyte was composed of 1 M $LiPF_6$ in a 1:1:1 of ethylene carbonate (EC), diethyl carbonate (DEC), and dimethyl carbonate (DMC). Charge-discharge measurements were performed on a LAND CT2001A multichannel battery testing system within a voltage window of 0.05–3.0 V. Cyclic voltammetry (CV) measurements were carried out on a CHI 660E electrochemical workstation at a sweep rate of 0.2 mV s$^{-1}$ within a voltage range of 0.0–3.0 V.

## Data availability
Data supporting the findings of this study are available within the article and the associated Supplementary Information Section. Any other data are available from the corresponding authors upon reasonable request. The source data underlying Fig. 3 and Supplementary Figs. are provided as a Source Data file.

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

## Acknowledgements

This work was supported by the National Key R&D Program of China (2018YFA0209401, 2018YFE0201701, and 2017YFA0207303), and the National Natural Science Foundation of China (21975050 and 21733003), the Key Basic Research Program of Science and Technology Commission of Shanghai Municipality (17JC1400100).

## Author contributions

D.Z., W.L., and Y.X. contributed to the conception and design of the experiments, analysis of the data and writing the manuscript. T.Z., X.Z., Y.Z., H.H., C.H., X.Z., Y.C., X.T., and J.W. assisted Y.X. for the synthesis of materials and the data collection and analysis. All authors contributed to the discussion and manuscript preparation.

## Competing interests

The authors declare no competing interests.
