## [Peer Review File · Nature Communications]

Reviewer #1 (Remarks to the Author):

The authors demonstrate a novel inorganic-organic competitive coating strategy for constructing gradient-structured Fe₃O₄@C nanospheres, in which the deposition of Fe₃O₄ nanoparticles and polymerization of carbonaceous species are competitive and well controlled by the reaction thermodynamics. The manuscript is well organized, so I would like to ask the authors to revise the following minor points. This work collects excellent data on the target subjects.

How about the materials' stability? This point is very critical if we consider practical applications.

The authors show wide-angle XRD patterns for samples. How about the average crystallite sizes? This size is matched with TEM data?

Related papers have been reported by different research groups. It is better to cite the following refs to support some related paragraphs in the introduction part (For example, Nano Energy, 10, 53-62, 2014, 53; ACS Nano, 2019, 138, 9607, etc.)

Overall the manuscript is well written, but I want to see the authors' perspective (future vision) on this research in the conclusion part.

Although this work may bring some new materials, I think the mechanism behind this material synthesis is not clear enough. It is better to explain the mechanism more carefully.

Reviewer #2 (Remarks to the Author):

This manuscript described the preparation of hollow gradient-structure of Fe₃O₄@C nanospheres strategies for Li battery anode. The hollow gradient-structure of Fe₃O₄@C which relieved the volume change of anode active materials during cycling resulted in long term cycle life of the battery.

- Although the hollow gradient-structure of Fe₃O₄@C showed stable long battery cycle life, the novelty of concept and material was not sufficiently enough. The several types of conversion-based anode materials have been investigated, including Fe₃O₄ materials. Moreover, the hollow structure strategy is also well-known in LIB anode materials. A comparative study with previous works should be comprehensively conducted.
- The absolute volume of Fe₃O₄@C nanosphere seems to be smaller than an original volume of SiO₂ template. The larger voids in the anode materials results in the loss of volumetric energy density. This issue should be addressed
- The areal mass loading was 1 mg/cm². This value is very low compared to practical ones in LIB anodes. Anodes with high mass loading should be prepared and their electrochemical performance should be provided.
- The composition ratio of Fe₃O₄/Carbon in the nanosphere should be provided
- The structural change of the hollow structure after 10000 cycles should be provided to demonstrate its structural stability.
- In-depth analysis of SEI layers on the anode materials should be conducted during cycling test.

Reviewer #3 (Remarks to the Author):

In their manuscript entitled "Inorganic-organic competitive coating strategy derived uniform hollow gradient-structured Fe₃O₄@C nanospheres for ultra-fast and long-term lithium-ion battery", Xia and coworkers report the synthesis of hollow carbon nanospheres with magnetite particles embedded in a gradient into the carbon shell along with the structure-performance relationships of these materials when applied as electrode materials in LIB anodes. The synthetic approach used here is very elegant in my view and provides a suitable way for the synthesis of such model materials. Authors give a conclusive investigation and discussion of the structure-property relationships of such materials in the electrochemical application which is surely helpful for the community. As a negative point I would like to point out that authors tend to "oversell" their results a bit ("stable robust electronic connections" and more) and also the language/typos and scientific expressions can still be improved ("iron ions cannot fully be escaped from coordination" and more). However, in general I believe that this work makes an outstanding example for the combination of materials synthesis with electrochemistry and deserves space in Nature Communications. Before possible publication, I suggest the following considerations.

- 1) Raman spectroscopy of the materials would be enormously helpful to get additional insights into the carbon microstructure and oxidation state of iron (magnetite and hematite can often not be easily distinguished in XRD, alternatively, Mössbauer spectroscopy would be helpful here). It is also not completely clear what the role and exact structure of the thin carbon layers in the electrochemical operation really is.
- 2) Porosity of such "closed" materials can eventually be better accessed with CO₂ physisorption. What is "mesoporous structure with few micropores". Authors should be more precise in this discussion.
- 3) It can not be concluded that the carbon content is 21% in case the iron gets further oxidized in TGA. It is likely to be at least in part Fe₂O₃. Also, a TGA measurement of the sample after etching by HF would be useful to check for the presence of remaining impurities.
- 4) For the electrochemical measurements authors apply current densities that are quite high for battery applications and are rather typical for Li-ion capacitors. I suggest to test the Lithium storage capacities also at lower C-rates to conclude more profound relationships between structure and properties.

REVIEWER COMMENTS

Reviewer #1 (Remarks to the Author):

The authors demonstrate a novel inorganic-organic competitive coating strategy for constructing gradient-structured Fe₃O₄@C nanospheres, in which the deposition of Fe₃O₄ nanoparticles and polymerization of carbonaceous species are competitive and well controlled by the reaction thermodynamics. The manuscript is well organized, so I would like to ask the authors to revise the following minor points. This work collects excellent data on the target subjects.

Response: We appreciate the reviewer's positive comments.

1. How about the materials' stability? This point is very critical if we consider practical applications.

Response: We thank the reviewer for the useful comments. Our gradient-structured Fe₃O₄@C nanospheres show excellent structural and thermal stability. The thermal stability of gradient-structured Fe₃O₄@C nanospheres is sufficient for practical applications. The electrodes were first undergone a hot- (90 °C, 30 Mpa) and cold-pressing (30 °C, 50 Mpa) process, then withstood a drying process of 80 °C under air atmosphere for 12 hrs. After these processes, the electrodes were assembled into batteries, which showed good performances. We clearly observed that all of the gradient-structured Fe₃O₄@C nanospheres were retained intact very well in the electrode after charge and discharge cycles. Those results show solid evidence that the gradient-structure has excellent mechanical and structural stability and is suitable for practical application.

Accordingly, we added the related description in Page 12 as follows:

“All of the gradient-structured Fe₃O₄@C nanospheres were retained intact very well on the surface of the electrode, indicating the excellent structural and thermal stability of gradient-structured Fe₃O₄@C nanospheres.”

We also added the detailed fabrication process of the electrode in Page 7 in revised Supplementary Information as follows:

“The mixing slurry was homogeneously pasted on nickel foam and dried in a vacuum oven at 80 °C for 12 h. After a hot- (90 °C, 30 Mpa) and cold-pressing (30 °C,

50 Mpa), the as-prepared electrodes were used to assemble the battery.”

We also revised the following caption in Page 35 in revised Supplementary Information as follows:

Fig. S26. *Ex-situ* SEM images of the hollow gradient-structured Fe₃O₄@C electrode and typical nanospheres. (A, D) the fresh one; (B, E) after 100 cycles at 10 Ag⁻¹; (C, F) after 1000 cycles at 10 Ag⁻¹.

2. The authors show wide-angle XRD patterns for samples. How about the average crystallite sizes? This size is matched with TEM data?

Response: We appreciate the reviewer very much for the useful comments. We have accepted the suggestion. The crystallite size of Fe₃O₄ nanoparticles is calculated by the Debye-Scherrer formula to be ~5.3 nm. This value is well-matched with TEM data (4-8 nm).

Accordingly, we added the related description in Page 7 as follows:

“The average crystallite sizes of the Fe₃O₄ can be calculated to ~5.3 nm by Debye-Scherrer formula. This value is well-matched with the TEM data of (4-8 nm).

We also added the following caption in Page 12 in revised Supplementary Information as follows:

Fig. S5. Superficial characteristics of the hollow gradient-structured Fe₃O₄@C nanospheres. (A) XRD pattern; (B) Raman spectra; (C) TGA curve; (D) nitrogen adsorption-desorption isotherms; (E) Adsorption curve of carbon dioxide; and (F) the pore size distribution.

3. Related papers have been reported by different research groups. It is better to cite the following refs to support some related paragraphs in the introduction part (For example, Nano Energy, 10, 53-62, 2014, 53; ACS Nano, 2019, 138, 9607, etc.)

Response: We have accepted the useful suggestions of the reviewer, the related papers have been cited as Reference No. 34 and 39 in the revised version. The information details of literature are as follows:

[34] S. Hwang, Y. Lim, J. Kim, Y. Heo, J. Lim, Y. Yamauchi, M. Park, Y. Kim, S. Dou,

- J. Kim, A case study on fibrous porous SnO₂ anode for robust, high-capacity lithium-ion batteries. *Nano Energy*, **10**, 53-62 (2014);
- [39] J. Lee, J. Moon, S. Han, J. Kim, V. Malgras, Y. Heo, H. Kim, S. Lee, H. Liu, S. Dou, Y. Yamauchi, M. Park, and J. Kim, Everlasting Living and Breathing Gyroid 3D Network in Si@SiO_x/C Nanoarchitecture for Lithium Ion Battery. *ACS Nano*, **13**, 9607-9619 (2019).

4. Overall the manuscript is well written, but I want to see the authors' perspective (future vision) on this research in the conclusion part.

Response: We appreciate the reviewer very much for the helpful comments. **We have accepted it and added the perspective (future vision) in the conclusion part at page 19 as follows:**

“We believe that our gradient-structured materials would bring a revolution in the field of ultrafast charge and discharge for lithium-ion batteries. The inorganic-organic competitive coating strategy paves a way for the design and synthesis of new core-shell and hollow structured materials for energy storage.”

5. Although this work may bring some new materials, I think the mechanism behind this material synthesis is not clear enough. It is better to explain the mechanism more carefully.

Response: We appreciate the reviewer for the comments. The novel gradient-structured Fe₃O₄@C shells are formed through an inorganic-organic competitive coating and deposition process, in which metal-organic ferrocene as the only source leads to the competitive deposition reactions of Fe oxides and amorphous carbon layer.

We have accepted the useful suggestions. In the revised manuscript, we gave an exhaustive description on the mechanism of the gradient-structure formation in page 14 as follows:

“We propose an inorganic-organic competitive coating and deposition process for the formation of the gradient-structured Fe₃O₄@C shells. At the beginning of the solvothermal reaction, the metallorganic ferrocene can be gradually hydrolyzed into

iron ions and carbonaceous specie by H_2O_2 . Then, the iron ions would be further hydrolyzed into Fe oxides and carbonaceous species could be polymerized into an amorphous carbon layer under the solvothermal condition. Thus, two competitive deposition reactions are occurred and much dependent on the thermodynamics for the controllable growth of unique gradient-structured $\text{Fe}_3\text{O}_4@\text{C}$ shells.”

REVIEWER COMMENTS

Reviewer #2 (Remarks to the Author):

This manuscript described the preparation of hollow gradient-structure of $\text{Fe}_3\text{O}_4@\text{C}$ nanospheres strategies for Li battery anode. The hollow gradient-structure of $\text{Fe}_3\text{O}_4@\text{C}$ which relieved the volume change of anode active materials during cycling resulted in long term cycle life of the battery.

Response: We appreciate the reviewer’s positive comments.

1. Although the hollow gradient-structure of $\text{Fe}_3\text{O}_4@\text{C}$ showed stable long battery cycle life, the novelty of concept and material was not sufficiently enough. The several types of conversion-based anode materials have been investigated, including Fe_3O_4 materials. Moreover, the hollow structure strategy is also well-known in LIB anode materials. A comparative study with previous works should be comprehensively conducted.

Response: We appreciate the reviewer for the comments. Conventional Fe_3O_4 -based anode materials suffer from drastic volume change. Various nanostructure designs, including yolk-shell, core-shell, *et al*, have been proposed to tackle this problem, but the results are still unsatisfactory. Gradient-structure, which can gradually release the stress caused by the volume change, is an ideal structure for ultra-fast charging and discharging.

The key innovation of this work lies in three aspects: (1) For the first time, we have designed and successfully prepared a unique hollow gradient-structured $\text{Fe}_3\text{O}_4@\text{C}$ nanosphere. (2) A novel inorganic-organic competitive coating strategy is demonstrated to synthesize gradient-structures. (3) Due to the gradient-structure

which can effectively relieve the stress concentration caused by drastic volume change, the anode demonstrates an outstanding specific capacity and excellent stability during ultra-fast charging and discharging.

We have revised the manuscript by adding comparative studies between the conventional hollow structure and our novel gradient structure.

We revised the description in Page 12 as follows:

“In fact, the cycle life and capacities of the gradient-structured $\text{Fe}_3\text{O}_4@\text{C}$ nanospheres are much better than that of the previous works under ultrafast charge and discharge conditions. Importantly, the gradient-structured $\text{Fe}_3\text{O}_4@\text{C}$ electrode can stand stable for a long term under fast charging and discharging. After 10000 cycles, it can deliver capacity as high as $\sim 750 \text{ mAh g}^{-1}$ with a Coulombic efficiency closed to 99.0% at 10 A g^{-1} (Fig. 3E).”

We also added the Table S1 as the comparison of the gradient-structured $\text{Fe}_3\text{O}_4@\text{C}$ nanospheres with previous work in page 40 in the revised Supporting Information.

2-1. The absolute volume of $\text{Fe}_3\text{O}_4@\text{C}$ nanosphere seems to be smaller than an original volume of SiO_2 template.

Response: We appreciate the reviewer for the comments. The scale bar of the $\text{Fe}_3\text{O}_4@\text{C}$ nanosphere image is smaller than that of the SiO_2 template image (Fig. S1). As a result, $\text{Fe}_3\text{O}_4@\text{C}$ nanosphere seems smaller than the SiO_2 template.

We have re-illustrated the figures so that they have a scale bar of the same length to avoid misunderstandings.

We revised Figure S1 and added the following figure caption in the revised Supporting Information:

Fig. S1. Morphology characterization of the pure colloidal SiO_2 cores and as-made gradient-structured nanospheres ($\text{SiO}_2@\text{G-Fe}_3\text{O}_4@\text{C}$) after the inorganic-organic competitive coating strategy at $210 \text{ }^\circ\text{C}$ under the solvothermal condition. (A, B) SEM images of the colloidal SiO_2 nanospheres with a particle size of 350 nm ; (C, D) SEM images of the as-made gradient-structured $\text{SiO}_2@\text{G-Fe}_3\text{O}_4@\text{C}$ nanospheres with a particle size of 420 nm .

2-2. *The larger voids in the anode materials results in the loss of volumetric energy density. This issue should be addressed.*

Response: We appreciate the reviewer for the comments. The larger voids of 350 nm in our hollow gradient-structure of Fe₃O₄@C nanospheres can be used not only as an electrolyte storage medium, but also for relieving the stress concentration caused by drastic volume change. Therefore, although a certain amount of volumetric energy density is sacrificed, the mass energy density is improved qualitatively. In the future, we will adjust the size of hollow voids to get the optimized volumetric energy density and ultra-fast, long-term stability performance.

Accordingly, we revised the description in Page 16 as follows:

“Moreover, the gradient-structure gradually relies on the void space to alleviate the volume change in the following discharging process. About 7% of the diameter increase for the gradient-structured Fe₃O₄@C nanospheres can be confirmed in comparison to the full charge and full discharge state, which is similar to the expansion coefficient of graphite anodes (6%), and doesn't cause the burst of SEI film on the surface. Although a certain amount of volumetric energy density is sacrificed, the mass energy density is improved qualitatively.”

3. *The areal mass loading was 1 mg/cm². This value is very low compared to practical ones in LIB anodes. Anodes with high mass loading should be prepared and their electrochemical performance should be provided.*

Response: We appreciate the reviewer for the comments. We agree and have accepted the suggestion of the reviewer. In the revised manuscript, we also tried different areal mass loading for battery performance research. In the case of a large loading, the preparation process determines the performance of the battery. With the increasing of the mass loading to 5, 10, and 20 mg/cm², the capacities of gradient-structure of Fe₃O₄@C nanospheres decrease to 727, 635 and 590 mAh g⁻¹, respectively, but their cyclic stabilities are maintained well enough to meet the requirement for its application.

Accordingly, we revised the description in Page 14 as follows:

“On the other hand, the cyclic stabilities of the gradient-structured Fe₃O₄@C can be

maintained well with a high loading density at 10 A g^{-1} . With the increasing the mass loading to 5, 10, and 20 mg/cm^2 , the capacities of gradient-structure of $\text{Fe}_3\text{O}_4@\text{C}$ nanospheres decrease to 727, 635 and 590 mAh g^{-1} , respectively (Fig. S30).”

4. The composition ratio of Fe_3O_4 /Carbon in the nanospheres should be provided.

Response: We greatly appreciate the reviewer for the comments. We accepted the suggestion of the reviewer. In the revised manuscript, the composition ratio of Fe_3O_4 /Carbon in the nanospheres was estimated to be about 76.2:23.8 (mass ratio) by the TGA measurements. In this work, all capacity values reported are based on the total mass of Fe_3O_4 and C in the nanospheres.

Accordingly, we revised the description in Page 7 as follows:

“The carbon content in the gradient-structured $\text{Fe}_3\text{O}_4@\text{C}$ nanospheres can be estimated to be about 23.8 % based on the TGA data (Fig. S5C, S6 and S7).”

We also added Figure S5, S6 and S7 in the revised Supporting Information, and the figure captions as follows:

Fig. S5. Superficial characteristics of the hollow gradient-structured $\text{Fe}_3\text{O}_4@\text{C}$ nanospheres. (A) XRD pattern; (B) Raman spectra; (C) TGA curve; (D) nitrogen adsorption-desorption isotherms; (E) Adsorption curve of carbon dioxide; and (F) the pore size distribution.

Fig. S6. The TGA curve of the sample after etching by HF to the removal of Fe_3O_4 . The result shows no presence of remaining impurities in the gradient-structured $\text{Fe}_3\text{O}_4@\text{C}$ nanospheres.

Fig. S7. The XRD pattern of the red product after TG analysis process. XRD result shows that all diffraction peaks can be well assigned to pure Fe_2O_3 (JCPDS card no.39-1346).

5. The structural change of the hollow structure after 10000 cycles should be provided to demonstrate its structural stability.

Response: We greatly appreciate the reviewer for the comments. We accepted the suggestion of the reviewer. We have carried out the relevant tests. The results show that the hollow gradient-structure is basically retained even after 10000 cycles at 10 A

g^{-1} . This result shows that the hollow gradient-structured $\text{Fe}_3\text{O}_4@\text{C}$ nanospheres have long-term stability as the anode for lithium ion battery.

Accordingly, we added the TEM images of the hollow structure after 10000 cycles (Fig. S22, S23) and revised the description in Page 11 as follows:

“Notably, the well-designed hollow gradient-structure can be perfectly retained at different states of charging and discharging, even after 100 or 10000 cycles at the different current densities of 0.2 A g^{-1} or 10 A g^{-1} (Fig. S22, S23).”

We also added the figure captions in Page 31 and 32 in the revised Supporting Information as follows:

Fig. S22. *Ex-situ* TEM images of the hollow gradient-structured $\text{Fe}_3\text{O}_4@\text{C}$ (HG- $\text{Fe}_3\text{O}_4@\text{C}$) nanospheres at a current density of 0.2 A g^{-1} . (A) After 30 cycles; and (B) after 100 cycles.

Fig. S23. *Ex-situ* TEM images of the gradient-structure of $\text{Fe}_3\text{O}_4@\text{C}$ nanospheres after 10000 cycles at 10 A g^{-1} . (A) Several nanospheres; and (B) Typical nanospheres.

6. In-depth analysis of SEI layers on the anode materials should be conducted during cycling test.

Response: We greatly appreciate the reviewer for the comments. We have accepted the suggestion of the reviewer. The growth of SEI films on the surface of the gradient-structured $\text{Fe}_3\text{O}_4@\text{C}$ nanospheres was investigated by scanning electron microscopy (SEM). The surface of the ball without discharge is relatively smooth. After 100 cycles at 10 A g^{-1} , the formation of SEI films makes the surface of the nanospheres coarser. The formation of some particles can be observed. After 1000 cycles at 10 A g^{-1} , the SEI film becomes thicker, so that the whole electrode surface is covered. Some pores between the nanospheres are also filled with SEI films. In fact, these SEI films are decomposed by high-energy electron beams when we detected the surface of gradient-structured $\text{Fe}_3\text{O}_4@\text{C}$ nanospheres with non-optical projections of TEM. Only smooth gradient-structured $\text{Fe}_3\text{O}_4@\text{C}$ nanospheres without SEI films can be observed (Fig. S26). This indicates that the SEI membrane is composed of some polymers similar to oligomers

Accordingly, we revised the description in Page 17 as follows:

“Moreover, the inner active layer in the gradient-structure is encapsulated by an out

amorphous carbon inactive layer, which enables SEI deposited on the outer surface of whole structures, thus spatially confined SEI formation (Fig. S26).”

We also added the new Figure S26 and its caption in Page 35 in the revised Supporting Information as follows:

Fig. S26. *Ex-situ* SEM images of the hollow gradient-structured Fe₃O₄@C electrode and typical nanospheres. (A, D) the fresh one; (B, E) after 100 cycles at 10 Ag⁻¹; (C, F) after 1000 cycles at 10 Ag⁻¹.

REVIEWER COMMENTS

Reviewer #3 (Remarks to the Author):

In their manuscript entitled "Inorganic-organic competitive coating strategy derived uniform hollow gradient-structured Fe₃O₄@C nanospheres for ultra-fast and long-term lithium-ion battery", Xia and coworkers report the synthesis of hollow carbon nanospheres with magnetite particles embedded in a gradient into the carbon shell along with the structure-performance relationships of these materials when applied as electrode materials in LIB anodes. The synthetic approach used here is very elegant in my view and provides a suitable way for the synthesis of such model materials. Authors give a conclusive investigation and discussion of the structure-property relationships of such materials in the electrochemical application which is surely helpful for the community. As a negative point I would like to point out that authors tend to "oversell" their results a bit ("stable robust electronic connections" and more) and also the language/typos and scientific expressions can still be improved ("iron ions cannot fully be escaped from coordination" and more). However, in general I believe that this work makes an outstanding example for the combination of materials synthesis with electrochemistry and deserves space in Nature Communications. Before possible publication, I suggest the following considerations.

Response: We appreciate the reviewer’s positive comments. We have accepted the suggestions of the reviewer. The language/typos and scientific expressions have been further improved in the revised version

1. Raman spectroscopy of the materials would be enormously helpful to get additional

insights into the carbon microstructure and oxidation state of iron (magnetite and hematite can often not be easily distinguished in XRD, alternatively, Mössbauer spectroscopy would be helpful here). It is also not completely clear what the role and exact structure of the thin carbon layers in the electrochemical operation really is.

Response: We greatly appreciate the reviewer for the comments. We have accepted the suggestions of the reviewer.

We have determined the carbon microstructure by the Raman spectra (Fig. S5B). They exhibit two bands at approximately 1345 and 1613 cm^{-1} , which are characteristic of the breathing mode of aromatic rings (the D band) and the bond stretching of the sp^2 carbon (the G band), respectively. The G band of the sample shifts to a higher wave number of 1613 cm^{-1} compared with the standard wave number of graphite single crystal (1575 cm^{-1}), indicating that a large amount of disordered graphite-like carbon exists in the gradient-structured $\text{Fe}_3\text{O}_4@\text{C}$ nanospheres.

The oxidation state of iron can be further determined by XRD pattern of the sample by removing the outer amorphous carbonaceous layers (Fig. S19). X-ray diffraction (XRD) peaks of the hollow hybrid $\text{Fe}_3\text{O}_4/\text{C}$ nanospheres can be well-assigned magnetite with $Fd3m$ symmetry (JCPDS: NO.11-0614). This result proves that the oxidation state of iron is magnetite once again.

In the gradient-structured $\text{Fe}_3\text{O}_4@\text{C}$ nanospheres, Fe_3O_4 nanoparticles are conformably encapsulated by the ultrathin graphitic carbon layer, which further enables the high electronic conductivity and ionic permeability through defects so that all nanoparticles are electrochemically active and mechanically stable. Furthermore, the ultrathin graphitic carbon layer ensures active particles independent of each other, thus prevents agglomeration of secondary particles during the charge and discharge cycling. As shown, even in full discharge state or after multiple cycles, Fe_3O_4 particles retain the initial encapsulated structure.

Accordingly, we added the Raman characterization of the materials as Figure S5B and the related description in Page 7 as follows:

“The G band of the carbon in the Raman spectra (Fig. S5B) shift to a higher wave number of 1613 cm^{-1} compared with that of the graphite single crystal (1575 cm^{-1}),

indicating that a large amount of disordered graphite-like carbon in the gradient-structured Fe₃O₄@C nanospheres.”

We also added the according figure caption in Page 35 in the revised Supporting Information as follows:

Fig. S5. Superficial characteristics of the hollow gradient-structured Fe₃O₄@C nanospheres. (A) XRD pattern; (B) Raman spectra; (C) TGA curve; (D) nitrogen adsorption-desorption isotherms; (E) Adsorption curve of carbon dioxide; and (F) the pore size distribution.

We also added the XRD characterization of the sample without amorphous carbonaceous layers as new Figure S19 and the related description in Page 28 in revised Supplementary Information as follows:

“X-ray diffraction (XRD) peaks of the hollow hybrid Fe₃O₄/C and yolk-shell structured Fe₃O₄@C nanospheres can be well-assigned magnetite with *Fd3m* symmetry (JCPDS: NO.11-0614).”

We also added the caption of Figure S19 in revised Supplementary Information as follows:

Fig. S19. Two kinds of commonly used nanostructures for the anode. (A) TEM images of the yolk-shell structured Fe₃O₄@C (YS-Fe₃O₄@C) nanospheres; (B) TEM images of the hollow hybrid Fe₃O₄/C (HH-Fe₃O₄/C); and (C) XRD patterns of the yolk-shell structured Fe₃O₄@C and hybrid hollow Fe₃O₄@C nanospheres. X-ray diffraction (XRD) peaks of the hollow hybrid Fe₃O₄/C and yolk-shell structured Fe₃O₄@C nanospheres can be well-assigned magnetite with *Fd3m* symmetry (JCPDS: NO.11-0614).

We also revised the description in Page 17 as follows:

“Furthermore, the ultrathin graphitic carbon layer ensures active particles independent of each other, thus prevents agglomeration of secondary particles during the charge and discharge cycling. As shown, even in full discharge state or after multiple cycles, Fe₃O₄ particles retain the initial encapsulated structure.”

We added the new Figure S21 and its caption in revised Supplementary Information as follows:

Fig. S21 *Ex-situ* TEM images of the hollow gradient-structured Fe₃O₄@C nanospheres at different states of charge and discharge during the 5th cycle at a current density of 0.2 A g⁻¹. (A) Without discharge; (B) half discharge; (C) full discharge, (D) half charge; (E) full charge; and (F) corresponding discharge and charge curves.

2. Porosity of such "closed" materials can eventually be better accessed with CO₂ physisorption. What is "mesoporous structure with few micropores". Authors should be more precise in this discussion.

Response: We appreciate the suggestions of the reviewer, we accepted them. We have determined the micropore volume from the Dubinin-Radushkevich (DR) equation with both N₂ adsorption at 77 K and CO₂ adsorption at 273 K (Fig. 5E). The obtained micropore volumes are very small compared with the total pore volume.

Accordingly, we added the CO₂ physisorption of the materials as Figure S5E and the related description in Page 7 as follows:

“The N₂ adsorption/desorption isotherms of the gradient-structured Fe₃O₄@C nanospheres show a typical type III feature (Fig. S5D), indicating that the microporosity in the gradient-structured Fe₃O₄@C nanospheres is non-significant. The micropore volumes determined from N₂ adsorption and CO₂ adsorption (Fig. S5E) are 0.005 and 0.007 cm³ g⁻¹, respectively, being negligible compared with the total pore volume (0.32 cm³ g⁻¹).”

We also added the caption of Figure S5 as follows:

Fig. S5. Superficial characteristics of the hollow gradient-structured Fe₃O₄@C nanospheres. (A) XRD pattern; (B) Raman spectra; (C) TGA curve; (D) nitrogen adsorption-desorption isotherms; (E) Adsorption curve of carbon dioxide; and (F) the pore size distribution.

3. It can not be concluded that the carbon content is 21% in case the iron gets further oxidized in TGA. It is likely to be at least in part Fe₂O₃. Also, a TGA measurement of the sample after etching by HF would be useful to check for the presence of remaining impurities.

Response: We greatly appreciate the reviewer for the comments. We agree with the reviewer that all the Fe_3O_4 could be got further oxidized to Fe_2O_3 during the TGA measurements. Actually, after the measurements, the sample is turned to red in color. Our XRD results show that the red product is Fe_2O_3 (Fig. S7). The total weight loss of the annealing gradient-structured $\text{Fe}_3\text{O}_4@\text{C}$ nanospheres is approximately 21.2%. 0.788 g of Fe_2O_3 could be obtained from 1.000 g of the gradient-structured $\text{Fe}_3\text{O}_4@\text{C}$ nanospheres according to TGA measurements. The Fe_3O_4 content of the gradient-structured $\text{Fe}_3\text{O}_4@\text{C}$ nanospheres is calculated to be about 76.2%. Consequently, the carbon content of the gradient-structured $\text{Fe}_3\text{O}_4@\text{C}$ nanospheres is about 23.8%. The TGA measurements of the sample after etching by HF were also carried out as shown in Fig. S6. The result shows no presence of remaining impurities.

We have accepted the suggestions of the reviewer. Accordingly, we revised the description in Page 14 as follows:

“The carbon content in the gradient-structured $\text{Fe}_3\text{O}_4@\text{C}$ nanospheres can be estimated to be about 23.8 % based on the TGA data (Fig. S5C, S6 and S7).”

We have added the new Figures S5, S6 and S7. Accordingly we have also added the figure caption as follows:

Fig. S5. Superficial characteristics of the hollow gradient-structured $\text{Fe}_3\text{O}_4@\text{C}$ nanospheres. (A) XRD pattern; (B) Raman spectra; (C) TGA curve; (D) nitrogen adsorption-desorption isotherms; (E) Adsorption curve of carbon dioxide; and (F) the pore size distribution.

Fig. S6. TGA curve of the sample after the removal of Fe_3O_4 by HF. The result shows no presence of remaining impurities in the gradient-structured $\text{Fe}_3\text{O}_4@\text{C}$ nanospheres.

Fig. S7. XRD pattern of the red product after TG analysis process. XRD result shows that all diffraction peaks can be well assigned to pure Fe_2O_3 (JCPDS card no.39-1346).

4. For the electrochemical measurements authors apply current densities that are quite high for battery applications and are rather typical for Li-ion capacitors. I suggest to test the Lithium storage capacities also at lower C-rates to conclude more profound relationships between structure and properties.

Response: We greatly appreciate the reviewer for the comments. We have accepted the suggestions of the reviewer. The lithium storage performances of the gradient-structured Fe₃O₄@C nanospheres were tested at a low C-rate of 0.2 A g⁻¹ (Fig. 3B). From the charge/discharge curves, it is clearly observed that the charging and discharging process show a voltage plateau near 1.6 and 0.9 V, respectively. This is characteristic of the conversion type anode materials for lithium ion batteries. The gradient-structured Fe₃O₄@C electrode shows an outstanding specific capacity of 1200 mAh g⁻¹ with a Coulombic efficiency of ~99.8% after 200 cycles, indicating that our materials possess excellent lithium storage performances also at lower C-rates.

Accordingly, we revised the description in Page 11 as follows:

“When charging and discharging in a low current density of 0.2 A g⁻¹ (Fig. 3B), the gradient-structured Fe₃O₄@C electrode demonstrates an outstanding specific capacity of 1200 mAh g⁻¹ with a Coulombic efficiency of ~99.8%, which is immensely higher than that of hollow island-type C@Fe₃O₄ (190 mAh g⁻¹) and hollow hybrid Fe₃O₄@C (990 mAh g⁻¹, 98.8%).

We also added modified Figure 3 and revised the caption as follows:

Fig. 3. Electrochemical characterizations of the hollow gradient-structured Fe₃O₄@C nanospheres (HG-Fe₃O₄@C) and the control samples application in lithium ion batteries. (A) Charge-discharge curves in the voltage range from 0.05 to 3.00 V; (B) Cycling performances at a current density of 0.2 A g⁻¹.

Reviewer #1 (Remarks to the Author):

The revised manuscript is highly improved.

Reviewer #2 (Remarks to the Author):

The revised manuscript has well addressed the reviewer's comments, along with the provision of experimental data. A following minor issue should be addressed.

- With increasing the mass loading (from 5 to 20 mg/cm²), the specific capacity of the nanospheres tended to decrease. The reason for this unusual behavior should be explained.

Reviewer #3 (Remarks to the Author):

The authors gave a complete and understandable response. The manuscript was revised according to my suggestions. I can, however, still not find additional electrochemical data at low current density. This is still suggested. Otherwise, the paper can be accepted from my point of view.

Manuscript Title: Inorganic-organic competitive coating strategy derived uniform hollow gradient-structured Fe₃O₄@C nanospheres for ultra-fast and long-term lithium-ion battery

Manuscript ID: NCOMMS-20-31164A

Point-by-Point Response to Referees:

Reviewer #1 (Remarks to the Author):

The revised manuscript is highly improved.

Response: We thank the reviewer very much for the approval of the work.

Reviewer #2 (Remarks to the Author):

The revised manuscript has well addressed the reviewer's comments, along with the provision of experimental data. A following minor issue should be addressed.

- With increasing the mass loading (from 5 to 20 mg/cm²), the specific capacity of the nanospheres tended to decrease. The reason for this unusual behavior should be explained.

Response: We thank the reviewer very much for the approval of the work. We appreciate the reviewer's suggestions. The thickness of the electrode was increased with the mass loading (from 5 to 20 mg/cm²), thus leading to the internal resistance increasing of whole electrode piece. Therefore, the specific capacity of the nanospheres were tended to decrease under ultrafast charge and discharge of 10 A/g.

We also explained the reason for this behavior in Page 14 in the revised manuscript as follow:

“With the increasing the mass loading to 5, 10, and 20 mg/cm², the capacities of gradient-structured Fe₃O₄@C nanospheres are decreasing to 727, 635 and 590 mAh g⁻¹, respectively, based on the thickness increasing of the electrode (Fig. S30).”

Reviewer #3 (Remarks to the Author):

The authors gave a complete and understandable response. The manuscript was revised according to my suggestions. I can, however, still not find additional electrochemical data at low current density. This is still suggested. Otherwise, the paper can be accepted from my point of view.

Response: We thank the reviewer very much for the approval of the work. We

appreciate the reviewer's suggestions. **We have added the electrochemical data at low current density of 0.1 C (1 C = 926 mA g⁻¹ based on Fe₃O₄) in Page S25 in the revised Supporting Information.**

Accordingly, we revised the description in Page 10 as follows:

“The lithium-storage characteristic of gradient-structured Fe₃O₄@C nanospheres were also tested at low current density of 0.1 C (1 C = 926 mA g⁻¹, Fig. S19).”